# Public Awareness Regarding Corticosteroid Use and Side Effects: A Cross-Sectional Study in Riyadh, Saudi Arabia

**DOI:** 10.3390/healthcare11202747

**Published:** 2023-10-16

**Authors:** Rayan Abubakker Qutob, Bassam Abdulaziz Alhusaini, Najd Khalid Aljarba, Omar Nasser Alzaid, Nawaf Abdulaziz Aljahili, Khalid Saad Alzahrani, Mohammed Mahmoud Sharaf, Abdullah Hussien Alghamdi, Abdullah Abdulaziz Alaryni, Yousef Mohammed Alammari, Abdulrahman Mohammed Alanazi, Fahad Ali Faqihi, Khalid Mohammed Al Harbi, Eysa Nahar Alsolamy, Osamah Ahmad Hakami

**Affiliations:** 1Department of Internal Medicine, College of Medicine, Imam Mohammad Ibn Saud Islamic University, Riyadh 11432, Saudi Arabia; dr.rayanq@hotmail.com (R.A.Q.); dr.alhomrani@gmail.com (A.H.A.); al3raini@hotmail.com (A.A.A.); yalammari@gmail.com (Y.M.A.); amn654@gmail.com (A.M.A.); khaharbi@yahoo.com (K.M.A.H.); eysa783@hotmail.com (E.N.A.); ohakami.md@gmail.com; 2Faculty of Medicine, Imam Mohammad Ibn Saud Islamic University, Riyadh 11432, Saudi Arabia; najddkhalid1999@gmail.com (N.K.A.); alzaidomar23@gmail.com (O.N.A.); nawafaljahili@gmail.com (N.A.A.); alzahranikds@gmail.com (K.S.A.); m.m.sharaf18@gmail.com (M.M.S.); 3Department of Adult Critical Care, Dr. Sulaiman Al Habib Medical Group, Riyadh 12231, Saudi Arabia; fahad.faqihi@drsulaimanalhabib.com

**Keywords:** awareness, corticosteroids, Saudi Arabia, side effects

## Abstract

The administration of corticosteroids may have possible hazards, ranging from minor adverse medication reactions to more serious considerations. We aimed to assess levels of public awareness concerning corticosteroid use, side effects, and predictors of its use. A cross-sectional study was conducted online throughout the period of May to July 2023. The present investigation utilized a previously developed questionnaire tool. The study encompassed a cohort of 732 individuals. Upon inquiry regarding the adverse effects of corticosteroid treatment, the participants predominantly reported weight gain, skin alterations, and fluid retention leading to breathing difficulties, constituting 44.4%, 30.3%, and 27.7% of the responses, respectively. The prevailing adverse effects observed in individuals using corticosteroids were weight gain, alterations in mood, and changes in skin characteristics, which accounted for 38.1%, 25.7%, and 21.8% of reported cases, respectively. Individuals within the age range of 41–50 years and those who are currently not working show a higher propensity for utilizing corticosteroids in comparison to other demographic groups (*p* < 0.05). The level of general public knowledge about corticosteroids and the side effects connected with them in Saudi Arabia was adequate. Demographic factors, such as age, gender, and education, have an impact on the use of corticosteroids.

## 1. Introduction

The adrenal cortex naturally produces corticosteroids, which have a variety of functions in the body [1,2]. Corticosteroids are categorized into two main classes according to their functions: mineralocorticoids and glucocorticoids [1,2]. Corticosteroids are among the most effective commonly used treatments for diverse autoimmune and inflammatory disorders [3]. These conditions include multiple sclerosis (MS); inflammatory bowel disease (IBD) (like ulcerative colitis and Crohn’s disease); painful and inflamed joints, tendons, and muscles; polymyalgia rheumatica; giant cell arteritis; urticaria (hives); hay fever; allergic rhinitis; chronic obstructive pulmonary disease (COPD); asthma; lupus; and atopic eczema [2].

Over time, it has become more evident that patients and physicians are misusing corticosteroids. In a previous study conducted in India, it was observed that a significant proportion of patients, specifically 88.9%, utilized potent and highly potent corticosteroid preparations based on recommendations from pharmacists, paramedical personnel, friends, and relatives [4]. Topical corticosteroids were recommended by physicians or dermatologists to only a limited number of individuals [4]. The primary motivation for using topical corticosteroids in 50.4% of patients, in the absence of any underlying dermatological condition, was to achieve skin lightening [4]. An additional 25.9% of patients used topical corticosteroids as a therapeutic intervention for the management of melasma and suntan [4]. Furthermore, dermatologists and general practitioners are increasingly prescribing corticosteroids to treat many illnesses, including melasma, urticarial, and undiagnosed skin rashes [5]. Due to the notable therapeutic effects of corticosteroids, physicians frequently prescribe them as a standard practice in order to attract patients to their clinics [6]. This practice typically entails the inappropriate utilization of corticosteroids for medical conditions when their use is not warranted [7].

A previous study in the United Arab Emirates (UAE) examined knowledge, attitude, and practice of the proper use of different dosage forms of corticosteroids among corticosteroids users and found poor knowledge, attitude, and practice results among different UAE patients using different corticosteroid dosage forms [8]. The utilization of corticosteroids may carry potential risks, varying from mild adverse drug reactions to more severe concerns, like immunosuppression and cardiovascular diseases [3,9]. These substantial health hazards are primarily associated with the prolonged systemic administration of elevated doses [3].

The degree of knowledge among patients is positively associated with optimal safety and clinical results. Accordingly, assessing the public comprehension of drugs and their proper usage is typically crucial for attaining the best therapeutic effect [10,11]. There is a shortage of population-based studies investigating attitudes and knowledge related to corticosteroids. These studies are notably restricted by their focus on specific dosage forms and small sample sizes [12]. In addition, another significant limitation arises from their exclusive concentration on particular demographic groups, like particular patient cohorts [13,14], caretakers [15,16,17,18], and/or healthcare professionals [19].

Given that corticosteroids are readily available and cost-effective treatment choices in developed nations, it is essential to assess the general awareness of these medications in such countries [20]. Thus, our study aimed to comprehensively assess levels of public awareness concerning corticosteroid use, side effects, and predictors of its use. We targeted both users and non-users in Riyadh, Saudi Arabia.

## 2. Methods

### 2.1. Study Design and Settings

The present study was a cross-sectional study conducted online throughout the period of May to July 2023. Its primary objective was to evaluate the extent of public knowledge pertaining to the utilization of corticosteroids and the associated side effects. The present study utilized a non-probability convenience sampling method. The co-authors conducted data collection through the utilization of an online-based questionnaire, which was disseminated via several social media platforms, including WhatsApp, Facebook, Instagram, and Snapchat. The hyperlink was shared on the primary webpage of the data collectors and social media pages, inviting anyone who satisfied the specified criteria for inclusion to engage in the research endeavor. In addition, participants were instructed to distribute the study link to their acquaintances and professional contacts.

### 2.2. Sample Population

The Saudi Arabian population living in Riyadh and older than 18 years of age formed the study population. There was no restriction on the participants’ gender or income category. Any participants who had mental issues or did not provide consent were excluded from the study.

### 2.3. Questionnaire Tool

The present investigation utilized a questionnaire tool created by Costello et al. to assess individuals’ perceptions of glucocorticoid side effects [21]. The individuals were questioned regarding their demographic characteristics in our questionnaire. Subsequently, an examination was conducted to assess the participants’ level of understanding of glucocorticoids and their associated side effects. The questionnaire assessed participants’ understanding of the various routes of administration and indications for corticosteroid use. It also inquired about their personal history of corticosteroid-based treatment, knowledge of associated side effects, and whether they received verbal or written information from their healthcare provider regarding the potential adverse effects of corticosteroid treatment. For those who reported prior corticosteroid use, the questionnaire further investigated whether they experienced any side effects during treatment. Lastly, participants were asked to identify their sources of information pertaining to corticosteroids.

### 2.4. Questionnaire Translation

The translation of the questionnaire into the Arabic language was conducted using the forward–backward approach. This approach involved the translator working independently to ensure that the original meaning of the questionnaire items was preserved, rather than relying on a literal word-for-word translation.

### 2.5. Face Validity Check and Piloting Phase

The clarity and comprehensibility of the questionnaire in its Arabic-translated version were assessed by experienced clinicians. The researchers verified that the items in the questionnaire are unambiguous and align with the aims of the study. Subsequently, a preliminary investigation was undertaken with a sample of individuals from the general population. The pilot study of the questionnaire confirmed several aspects, including the clarity and comprehensibility of the survey items, their alignment and suitability with the research objectives, the adequacy of the questionnaire’s length and completion time, and the appropriateness of the response options and scale employed.

### 2.6. Sample Size

According to the latest available statistics for 2022, the total population in Riyadh, Saudi Arabia, is around 7 million. Using a confidence interval of 95%, a standard deviation of 0.5, a power of 80%, and a margin of error of 5%, the required sample size was 385 participants. Raosoft sample size estimator was used to estimate the sample size for this study.

### 2.7. Statistical Analysis

Categorical variables were presented using frequency and percentage. A binary logistic regression analysis was used to identify predictors of corticosteroid use. The dependent variable in the regression model was defined as the use of corticosteroids and the independent variables were participants’ demographic characteristics. Chi-square test was used to compare proportions between users and non-users. A 95% confidence interval (*p* ≤ 0.05) was applied to indicate the statistical significance of the results, and a significance level of 5% was assigned. All data were analyzed using the Statistical Package for Social Science software (version 27).

### 2.8. Ethical Approval

This study was approved by the Institutional Review Board at Al-Imam Muhammad Ibn Saud Islamic University, Riyadh, Saudi Arabia (Project number: 487/2023). All patients gave their consent before taking part in this study.

## 3. Results

### 3.1. Participants’ Demographic Characteristics

A total of 732 participants were involved in this study. Around one-third of the study participants (32.2%) were aged 21–30 years. Half of the study sample (50.0%) were males. The vast majority of the study participants (92.5%) were Saudis. More than half of the study sample (60.9%) reported that they held a bachelor’s degree. Around one-third of the study sample (28.3%) were university students. Almost half of the study participants (49.7%) were single. Table 1, below, presents the demographic characteristics of the study participants.

There was a significant difference in the proportion of corticosteroid users and non-users in term of the their demogaphic characteristics such as age, gender, education, employment, and marital status (*p* < 0.05).

### 3.2. Knowledge of the Administration Method and Indications of Use

Table 2 presents knowledge of the administration method and indication of use among the study participants. When the participants were asked how corticosteroids could be given, the most commonly reported administration methods were orally, topically, and intravenous injection, accounting for 59.7%, 49.9%, and 43.6%, respectively. The most commonly reported indications for corticosteroid treatment were asthma, rheumatoid disease, and severe allergy, accounting for 49.6%, 47.4%, and 41.5%, respectively.

### 3.3. Community’s Awareness of Side Effects

Table 3 presents the community’s awareness of side effects. When the participants were asked about corticosteroid treatment side effects, the most commonly reported side effects were weight gain, skin changes (e.g., bruising, tender skin, stretch marks, redness of the face), and fluid accumulation in the body with difficulty breathing, accounting for 44.4%, 30.3%, and 27.7%, respectively.

### 3.4. Corticosteroid Users’ Awareness of Side Effects, and Evaluating the Doctor’s Level of Awareness

Table 4 presents corticosteroid users’ awareness of side effects and evaluates the doctor’s level of awareness. Around one-quarter of the study participants (27.6%) reported that they had received corticosteroid treatment before, of which 41.1% reported that they received verbal or written information from their doctor regarding the side effects of corticosteroid treatment. The most commonly reported side effects among corticosteroid users were weight gain, mood changes, and skin changes (e.g., bruising, thinning skin, stretch marks), accounting for 38.1%, 25.7%, and 21.8%, respectively.

### 3.5. Participants’ Sources of Information Regarding Corticosteroid Treatment

Table 5, below, presents participants’ sources of information regarding corticosteroid treatment. The most commonly reported sources of information regarding corticosteroid treatment were friends and family, healthcare providers, and social media, accounting for 45.5%, 38.8%, and 28.8%, respectively.

### 3.6. Predictors of Using Corticosteroids

Participants who are aged 41–50 years and unemployed were more likely to use corticosteroids compared to others (*p* < 0.05). On the other hand, males and those with a low level of education (intermediate school level or lower) were less likely to use corticosteroids compared to others (*p* < 0.05), Table 6.

## 4. Discussion

In this cross-sectional study, we assessed levels of awareness about corticosteroid use and associated side effects among users and non-users within our study population. The results of our research highlight significant points about knowledge and perceptions of corticosteroid use in a healthcare context.

The choice of corticosteroid administration method mainly depends on the treated condition. Corticosteroids can be administered through various routes, including oral, topical, intravenous, intramuscular, intraarticular, inhaled, etc. [22]. Allergies, asthma, inflammatory bowel disease (IBD), rheumatoid arthritis, and many other disorders are all treated with corticosteroid medications [3,23]. Our study found that the most frequently reported methods of corticosteroid administration among the participants were oral ingestion, topical application, and intravenous injection, accounting for 59.7%, 49.9%, and 43.6%, respectively. In addition, the most commonly reported indications for corticosteroid treatment were asthma, rheumatoid disease, and severe allergy, accounting for 49.6%, 47.4%, and 41.5%, respectively. These findings imply that study participants have a good knowledge of corticosteroid administration methods and common indications for administering corticosteroids. This awareness may be due to the participants’ educational level, with 60.9% holding bachelor’s degrees and 9.3% having completed higher education. In line with our findings, asthma was documented as the primary reason for oral corticosteroid administration, constituting 39.2% of cases in a previous hospital-based study [24]. Another prior study also found that the most commonly reported reason for oral corticosteroid administration was respiratory disease (40%) [25].

The awareness and knowledge of corticosteroids have been evaluated in numerous prior studies [26,27,28]. Aligning with our research findings, a nationwide survey conducted in South Korea identified a notable prevalence of incorrect responses related to the application methods and indications of corticosteroids in certain aspects. Nevertheless, respondents demonstrated satisfactory overall knowledge [27]. A prior multinational study also documented that participants have adequate knowledge about corticosteroid usage [29]. In India, there was a notable lack of awareness and knowledge concerning steroids, with approximately 83% of participants lacking familiarity with steroids and steroid-related information [28].

The most frequent adverse effects of corticosteroid administrations include Cushingoid features, cataracts and glaucoma, myopathy, dermatologic and gastrointestinal adverse effects, hyperglycemia and diabetes, immunosuppression, psychiatric disturbances, fractures, osteoporosis, cardiovascular disease, and hypothalamic–pituitary–adrenal (HPA) axis suppression [2,3]. When our study participants were asked about corticosteroid treatment side effects, the most commonly reported side effects were weight gain, skin changes (e.g., bruising, tender skin, stretch marks, and redness of the face), and fluid accumulation in the body with difficulty breathing, accounting for 44.4%, 30.3%, and 27.7%, respectively. These findings indicate that participants are aware of potential corticosteroid side effects. It is positive that participants are aware of these side effects, as they can significantly impact a patient’s quality of life. However, it is crucial to investigate whether this awareness influences treatment adherence and whether patients consult their healthcare providers about these concerns. In line with our results, a previous study was carried out to assess dentistry students’ knowledge concerning corticosteroids’ adverse effects. The findings indicated that most of these students demonstrated awareness of adverse effects [30].

In contrast to our results, a previous study revealed that King Fahad Specialist Hospital attendees demonstrated a limited understanding of the side effects associated with using corticosteroids. Furthermore, within this group, those who used corticosteroids were more informed about these side effects compared to non-users, and it was noted that the occurrence of the disease contributed to heightened awareness of these side effects [31]. Similarly, a prior multinational study also identified a low level of awareness among participants regarding corticosteroid side effects [29]. Likewise, comparable results were reported by a previous nationwide survey conducted in South Korea concerning the general public’s awareness of topical corticosteroids [27]. The apprehension of experiencing adverse effects represents a significant obstacle to steroid utilization. Consequently, healthcare providers play an essential role in counseling patients effectively about the proper application of corticosteroids, thus preventing the abuse, hesitation, and misuse of these medications.

Around one-quarter of our study participants (27.6%) reported prior corticosteroid treatment, with 41.1% of them indicating that they received information, either verbally or in writing, from their doctors regarding the potential side effects of corticosteroids treatment. Additionally, a previous investigation revealed that approximately 31.6% of the study participants had reported prior usage of corticosteroids, personally or by friends or family [29]. Within this subgroup of individuals who had previously used corticosteroids, a significant preponderance (36.7%) had utilized corticosteroid medications for a duration of fewer than seven days [29]. These findings highlight the crucial role of healthcare practitioners in educating individuals about the potential risks associated with corticosteroid use. The importance of improving communication between patients and healthcare providers becomes evident when considering that a notable proportion of corticosteroid users did not receive this information from their doctors. Therefore, to enhance the quality of care and ensure patient safety, it is necessary to ensure that all corticosteroid users are well-informed about the possible adverse effects.

In our study, among those using corticosteroids, the most commonly reported side effects were weight gain, mood changes, and skin changes (e.g., bruising, thinning skin, stretch marks), accounting for 38.1%, 25.7%, and 21.8%, respectively. These results are consistent with the documented side effects of corticosteroid use, as mentioned before. Again, recognizing common side effects among users highlighted the relevance of healthcare providers’ roles in informing patients about the potential adverse effects of corticosteroid medication and the need for continued monitoring and support to manage these side effects properly. Consistent with our results, participants reported multiple side effects in a previous study. The most common was heightened appetite, resulting in weight gain, in 45% of respondents. Mood alterations, mood swings, and depression followed closely behind, affecting 34.1% of individuals. Additionally, 27.8% of the individuals who utilized corticosteroids reported experiencing acne, and 26.1% had easily bruised, thin skin [29]. 

In our study, the most commonly reported sources of information regarding corticosteroid treatment were friends and family, healthcare providers, and social media, accounting for 45.5%, 38.8%, and 28.8%, respectively. While healthcare providers emerge as trusted sources, the reliance on friends and family underscores the influence of personal networks in obtaining knowledge about corticosteroids. Nevertheless, it is crucial to acknowledge that information obtained from friends and family may not always be accurate. Furthermore, the impact of social media raises concerns about the potential for misinformation [32,33]. These observations highlight the necessity for a comprehensive strategy, which includes educating healthcare professionals, promoting responsible engagement on social media platforms, and empowering individuals to critically evaluate corticosteroid-related information from diverse sources. Ultimately, such measures will contribute to a better understanding of corticosteroid therapy. In line with our findings, a prior study observed that individuals more frequently depend on friends and family for information in comparison to consulting physicians or pharmacists [27]. While friends and family may have some degree of proficient knowledge, they typically lack formal healthcare training, raising concerns about the potential for medication misuse or overuse [34]. On the contrary, earlier studies have indicated that physicians or pharmacists tend to be the primary sources of medication information within interpersonal networks [34,35,36,37]. Moreover, an earlier study revealed that social media sites were the primary source of information regarding corticosteroids for most of the participants [29].

In our study, corticosteroid use was more prevalent among participants aged 41–50 years and unemployed compared to others (*p* < 0.05). Conversely, males and individuals with low educational attainment (intermediate school or lower) were less likely to use corticosteroids compared to others (*p* < 0.05). These findings highlight the significance of considering variables such as age, employment status, gender, and educational background when investigating corticosteroid utilization patterns, indicating the necessity of specialized healthcare interventions and educational programs catered to various demographic groups to promote proper corticosteroid usage. A prior study demonstrated that the implementation of an educational module on corticosteroids led to a substantial increase in knowledge among participants [28].

The higher likelihood of using corticosteroid among participants aged 41–50 and unemployed may indicate that this demographic is more susceptible to medical conditions necessitating corticosteroid therapy. Furthermore, this finding suggests that improving access to medical care among this group could help reduce steroid phobia and ultimately increase corticosteroid usage among other patient groups who might need it for their health conditions. In a similar context, a prior study revealed that individuals experiencing long-term unemployment face a significantly higher health burden when compared to both employed individuals and those with shorter periods of unemployment [38]. This previous study confirmed a direct correlation between the duration of unemployment and the increased disease burden. A previous meta-analysis indicated that individuals without employment have approximately 30% more potential to utilize healthcare services [39].

A previous study reported a significant negative association between developing corticophobia and the participants’ knowledge and educational levels [29]. Health consequences are widely recognized to be significantly associated with sociodemographic factors, particularly educational level [40,41,42]. These may explain the lower probability of corticosteroid use among males and individuals with lower education levels among our study participants. Moreover, a prior study revealed that when analyzing the association between demographic data and awareness scores, it was observed that females scored higher in awareness, with a score of 53%, compared to males, who had an awareness score of 33% [43].

This study has limitations. The cross-sectional study design restricted our ability to examine causality between study variables. The generalizability of our study findings might have been affected as our study population was restricted to Riyadh city. In Riyadh, similar to several major cities, it is not atypical to observe a somewhat greater proportion of young people and professionals in comparison to other regions within the kingdom. The gender distribution of Riyadh may be subject to impact from the substantial expatriate community, particularly in industries such as construction and services, where male expats are frequently employed. Riyadh, being the capital and a prominent commercial hub, has a greater degree of cosmopolitanism and population diversity. The location draws individuals from several places within Saudi Arabia, as well as from numerous nations, seeking employment and entrepreneurial prospects.

## 5. Conclusions

The level of awareness among the general population in Saudi Arabia regarding corticosteroids and their associated adverse effects was found to be good. The consumption of corticosteroids is impacted by demographic variables such as age, gender, and education level. It is imperative for individuals to obtain accurate information regarding corticosteroids from credible sources, and it is strongly advised that they refrain from obtaining these medications without a valid prescription.

## Figures and Tables

**Table 1 healthcare-11-02747-t001:** Participants’ demographic characteristics.

Variable	Frequency	Percentage	Frequency	Percentage	Frequency	Percentage	*p*-Value
	Overall		Users		Non-Users		
**Age categories**							
18–20 years	106	14.5%	24	11.9%	82	15.5%	0.040
21–30 years	236	32.2%	55	27.2%	181	34.2%	
31–40 years	194	26.5%	53	26.2%	141	26.6%	
41–50 years	109	14.9%	43	21.3%	66	12.5%	
51–60 years	71	9.7%	22	10.9%	49	9.2%	
61 years and above	16	2.2%	5	2.5%	11	2.1%	
**Gender**							
Males	366	50.0%	74	36.6%	292	55.1%	<0.001
**Nationality**							
Saudi	677	92.5%	183	90.6%	494	93.2%	0.149
**Education**							
Primary school or lower	5	0.7%	5	2.5%	0	0.0%	0.005
Intermediate school	9	1.2%	4	2.0%	5	0.9%	
Secondary school	204	27.9%	45	22.6%	159	30.0%	
Bachelor’s degree	446	60.9%	121	60.8%	325	61.3%	
Higher education	68	9.3%	27	13.6%	41	7.7%	
**Employment**							
Student	207	28.3%	50	24.8%	157	29.6%	0.015
Employed (governmental sector)	172	23.5%	46	22.8%	88	16.6%	
Employed (private sector)	180	24.6%	15	7.4%	24	4.5%	
Unemployed	134	18.3%	37	18.3%	143	27.0%	
Retired	39	5.3%	54	26.7%	118	22.3%	
**Marital status**							
Married	336	45.9%	116	57.4%	220	41.5%	<0.001
Single	364	49.7%	75	37.1%	289	54.5%	
Divorced	25	3.4%	9	4.5%	16	3.0%	
Widowed	7	1.0%	2	1.0%	5	0.9%	

**Table 2 healthcare-11-02747-t002:** Knowledge of the administration method and indication of use.

Variable	Frequency	Percentage
**In your opinion, how can corticosteroid treatment be given?** (You can choose more than one option)		
Orally	437	59.7%
Topically	365	49.9%
Intravenous injection	319	43.6%
Inhalation	227	31.0%
Intramuscular injection	176	24.0%
Eye drops	164	22.4%
Articular injection	136	18.6%
Dermal injection	111	15.2%
Rectally	59	8.1%
**In your opinion, what are the cases that require corticosteroid treatment?** (You can choose more than one option)		
Asthma	363	49.6%
Rheumatoid disease	347	47.4%
Severe allergy	304	41.5%
Inflammation of the joints, muscles and tendons	295	40.3%
Lupus	204	27.9%
Chronic obstructive pulmonary disease	173	23.6%
Joint stiffness	161	22.0%
Ulcer	137	18.7%
Kidney stone	128	17.5%
Recurrent gastrointestinal infections (Crohn’s and ulcerative colitis)	105	14.3%
Blood cancer (leukaemia)	99	13.5%
Glomerulonephritis	98	13.4%
Adrenal insufficiency—Addison’s disease	86	11.7%
Lymphomas	85	11.6%
Cardiovascular diseases	77	10.5%
Diabetes mellitus	77	10.5%
Hypertension	72	9.8%
Mental illnesses (anxiety and depression)	62	8.5%
Dyslipidaemia	57	7.8%
Cushing’s syndrome	56	7.7%
Gastroesophageal reflux disease.	48	6.6%
Erectile dysfunction	43	5.9%

**Table 3 healthcare-11-02747-t003:** Community’s awareness of side effects.

Variable	Frequency	Percentage
**In your opinion, which of the following is a side effect of corticosteroid treatment?** (You can choose more than one option)		
Weight gain	325	44.4%
Skin changes (e.g., bruising, tender skin, stretch marks, redness of the face)	222	30.3%
Fluid accumulation in the body with difficulty breathing	203	27.7%
Acne/pimples on the face	202	27.6%
Mood changes	188	25.7%
Heart palpitations and heart attack	161	22.0%
Osteoporosis	160	21.9%
High blood sugar level	151	20.6%
Face roundness	142	19.4%
Eye diseases (e.g., white water, glaucoma)	133	18.2%
Depression	123	16.8%
Hair loss	117	16.0%
Hypertension	114	15.6%
Diarrhea	110	15.0%
Anorexia	108	14.8%
Palpitations	108	14.8%
Infection (e.g., pneumonia)	106	14.5%
Insomnia	96	13.1%
Constipation	85	11.6%
Indigestion	81	11.1%
Cold extremities	75	10.2%
Anemia	64	8.7%
Decreased daily urination	57	7.8%
Erectile dysfunction	48	6.6%
Memory loss	47	6.4%
Epilepsy	34	4.6%

**Table 4 healthcare-11-02747-t004:** Corticosteroid users’ awareness of side effects, and evaluating the doctor’s level of awareness.

Variable	Frequency	Percentage
**Have you ever taken corticosteroid treatment?**		
Yes	202	27.6%
**During your diagnosis/treatment, did you receive verbal or written information from your doctor regarding the side effects of corticosteroid treatment?** (n = 202)		
Yes	83	41.1%
No	63	31.2%
I don’t remember	56	27.7%
**Did you experience any of these side effects while taking corticosteroids?** (You can choose more than one option) (n = 202)		
Weight gain	77	38.1%
Mood changes	52	25.7%
Skin changes (e.g., bruising, thinning skin, stretch marks)	44	21.8%
Acne/pimples on the face	37	18.3%
palpitations	36	17.8%
Insomnia	31	15.3%
Face roundness (moon face)	31	15.3%
Indigestion	20	9.9%
Hypertension	18	8.9%
Osteoporosis	18	8.9%
Eye diseases (e.g., white water, glaucoma)	16	7.9%
High blood glucose level	14	6.9%
Infection (ex: pneumonia)	11	5.4%
Cardiovascular diseases (ex: heart attack)	9	4.5%

**Table 5 healthcare-11-02747-t005:** Participants’ sources of information regarding corticosteroid treatment.

Variable	Frequency	Percentage
**What are your sources of information about corticosteroid treatment?** (You can choose more than one option)		
Friends and family	333	45.5%
Healthcare providers	284	38.8%
Social media	211	28.8%
Medical websites	175	23.9%
Public websites	96	13.1%
Magazines and books	59	8.1%

**Table 6 healthcare-11-02747-t006:** Predictors of using corticosteroids.

**Variable**	**Odds Ratio of Using Corticosteroids (95% Confidence Interval)**	** *p* ** **-Value**
**Age categories**		
18–20 years (reference group)	1.00	
21–30 years	1.04 (0.60–1.79)	0.893
31–40 years	1.28 (0.74–2.23)	0.376
41–50 years	2.23 (1.23–4.04)	0.008
51–60 years	1.53 (0.78–3.02)	0.216
61 years and above	1.55 (0.49–4.91)	0.453
**Gender**		
Females (reference group)	1.00	
Males	0.47 (0.34–0.66)	<0.001
**Education**		
Higher education (reference group)	1.00	
Secondary school	-	-
Intermediate school	0.43 (0.24–0.77)	0.005
Bachelor’s degree	1.22 (0.30–4.93)	0.786
Primary school or lower	0.57 (0.33–0.96)	0.035
**Employment**		
Student (reference group)	1.00	
Unemployed	1.64 (1.02–2.65)	0.042
Retired	1.96 (0.96–4.03)	0.066
Employed (private sector)	0.81 (0.50–1.32)	0.398
Employed (governmental sector)	1.44 (0.91–2.26)	0.117
**Marital status**		
Widowed (reference group)	1.00	
Single	0.65 (0.12–3.41)	0.609
Married	1.32 (0.25–6.90)	0.744
Divorced	1.41 (0.23–8.78)	0.715

## Data Availability

All data are available in the manuscript.

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
