# Peer review of "Public Awareness Regarding Corticosteroid Use and Side Effects: A Cross-Sectional Study in Riyadh, Saudi Arabia"

_healthcare, 2023, doi:10.3390/healthcare11202747_

Round 1
Reviewer 1 Report
Thank you for conducting this important study and presenting the results. Please find my comments below.
Title:
1. I am having a hard time with the title especially after reviewing the article, especially since the results aren't separated between those who have used corticosteroids and those that don't.
2. I am also a bit concerned that you use cortisone and corticosteroids interchangeably both in the paper and in your questionnaire. I understand that cortisone is the most commonly used steroid in your country, but that is a bit deceiving when it comes to the title
Introduction
1. Line 57 - You state "Over time, it has become more evident that patients and physicians are misusing corticosteroids." There needs to be evidence as well as a description of what this misuse is
2. Line 71 - You discuss "general awareness of these medications" and yet there is no distinction between the general public and prescribers when it comes to the presentation of your results
Methods
1. Your study design is questionable - you sent out a link to random strangers on several social media platforms. How did you get their usernames, how did you confirm that these individuals were not lying about the information they presented, how did you find which individuals to send these links out to - was there any exclusion, and if so, how did you confirm that people were not lying about their age, etc.
2. Line 109 - You state that the questionnaire was assessed by experienced clinicians. I commend you for taking the time to appropriately translate the survey for readability and comprehension, however, many of your study participants had a low level of education - how did you ensure that your educated clinicians took health literacy into account?
3. When you discuss your statistical analysis, you state that you were looking to identify predictors of corticosteroid use, however there is no determination of power to determine how many participants you actually needed in order to see if there are predictors; Furthermore, with such a small sample population vs the population of Saudi Arabia, identifying predictors is questionable.
Results
1. As I mentioned previously, you initially started your paper discussing corticosteroids and then mentioned only cortisone in your questions. For an individual with very low health literacy, they may be confused and not understand the connection between the two.
2. Along the same lines, how did you ensure that individuals who had a low level of education understood difficult terms like "Glomerulonephritis" and "Cushing's Syndrome" - this may cause your results to be skewed
3. In Table 4, you asked: "Have you ever taken cortisone treatment (corticosteroid)?" Did you take this into account when it comes to your results? Perhaps folks who have taken the medication before are therefore more likely to know more about the medication than others, and so once again, this could skew your results.
4. Along these same lines, there is no delineation between the results as it comes to the general public vs. healthcare providers vs. what students were in university for, those who are furthering their education in the medical profession may have more knowledge.
5. Lines 171 - 175 - The section called Predictors of corticosteroids, please see my comments below; you cannot make this kind of claim when there are many other factors that need to be considered.
Discussion
1. Line 191 - You make the claim that "these findings imply that study participants have a good knowledge of corticosteroid administration methods..." What defines good? What are you comparing it to?
2. Line 195 - What is a chest disease?
3. Line 205 - You state "In contrast to our research findings, an earlier study found that pharmacists exhibited knowledge gaps regarding administration..." What does this have to do with your research findings when you never indicated that you had pharmacists in your cohort or that you looked specifically at pharmacist results?
4. Line 297 - You state "Furthermore, this finding suggests that improving access to medical care among this group could help reduce steroid phobia..." And yet there is no question regarding this in your questionnaire.
I appreciate the efforts that it took to complete this study - you had a large number of participants - however, there are several methodological issues that prevent me from recommending this paper to proceed.
N/A
Author Response
Reviewer 1:
Thank you for conducting this important study and presenting the results. Please find my comments below.
Title:
- I am having a hard time with the title especially after reviewing the article, especially since the results aren't separated between those who have used corticosteroids and those that don't.
- Thank you for this comment. We are sorry for this confusion, we have now re-wrote the title of the study to address the reviewer comment to be “Public awareness level regarding corticosteroid use and side effects: A cross-sectional study among user and non-user in Riyadh, Saudi Arabia”.
2. I am also a bit concerned that you use cortisone and corticosteroids interchangeably both in the paper and in your questionnaire. I understand that cortisone is the most commonly used steroid in your country, but that is a bit deceiving when it comes to the title.
- Thank you for this comment. We are sorry for this unintended writing mistake. In our Arabic translated version of the questionnaire we used the general term of corticosteroids as our study was not specified for cortisone treatment. We have now removed the word “cortisone” from the manuscript and replaced it with “corticosteroids”.
Introduction
- Line 57 - You state "Over time, it has become more evident that patients and physicians are misusing corticosteroids." There needs to be evidence as well as a description of what this misuse is.
- Thank you for this comment. We have now addressed this comment and clarified this statement further and defined misuse, see page 3, lines 72-85.
- Line 71 - You discuss "general awareness of these medications" and yet there is no distinction between the general public and prescribers when it comes to the presentation of your results
- Thank you for this comment. In the introduction, we provided an overall background information about the importance of this topic to be examined among different populations. This was done to engage the readers and give them a broader understanding about this topic. However, the aim of the study is clearly stated that we assessed public awareness concerning corticosteroids use and side effect, see page 4, lines 103-104.
Methods
- Your study design is questionable - you sent out a link to random strangers on several social media platforms. How did you get their usernames, how did you confirm that these individuals were not lying about the information they presented, how did you find which individuals to send these links out to - was there any exclusion, and if so, how did you confirm that people were not lying about their age, etc.
- Thank you for this comment. Using online survey study design is very common in our field. We call the employed sampling technique as convenience sampling technique which is non-random sampling technique. It is based on recruiting the participants based on their availability and willingness to participate. We shared the questionnaire link on multiple social media pages. The participants in each page were asked to participate after explaining the inclusion criteria for them and the study purpose. They were informed that the participation is voluntarily and we did not collect any personal identifiable information for them to let them participate with more comfort and make their participation more reliable. No one can confirm the information provided using online survey study design in any study. This is a common limitation. However, the questions that we presented in our study to the participants were not sensitive or personal, we asked them about their use of corticosteroids. In addition, before conducting the study, we conducted a small pilot study with a sample of individuals from the general population who confirmed that the questions are suitable, see pages 4 and 5.
- Line 109 - You state that the questionnaire was assessed by experienced clinicians. I commend you for taking the time to appropriately translate the survey for readability and comprehension, however, many of your study participants had a low level of education - how did you ensure that your educated clinicians took health literacy into account?
- Thank you for this comment. As we mentioned under the subheading “Face Validity Check and Piloting Phase”, a preliminary investigation was undertaken with a sample of individuals from the general population. The pilot study of the questionnaire confirmed several aspects, including the clarity and comprehensibility of the survey items, their alignment and suitability with the research objectives, the adequacy of the questionnaire's length and completion time, and the appropriateness of the response options and scale employed, see page 5, lines 138-146.
- When you discuss your statistical analysis, you state that you were looking to identify predictors of corticosteroid use, however there is no determination of power to determine how many participants you actually needed in order to see if there are predictors; Furthermore, with such a small sample population vs the population of Saudi Arabia, identifying predictors is questionable.
- Thank you for this comment. We have now added sample size calculation procedure under the sub-heading “sample size”, see page 5, lines 147-1451.
Results
- As I mentioned previously, you initially started your paper discussing corticosteroids and then mentioned only cortisone in your questions. For an individual with very low health literacy, they may be confused and not understand the connection between the two.
- Thank you for this comment. We are sorry for this unintended writing mistake. In our Arabic translated version of the questionnaire we used the general term of corticosteroids as our study was not specified for cortisone treatment. We have now removed the word “cortisone” from the manuscript and replaced it with “corticosteroids”.
- Along the same lines, how did you ensure that individuals who had a low level of education understood difficult terms like "Glomerulonephritis" and "Cushing's Syndrome" - this may cause your results to be skewed.
- Thank you for this comment. As we mentioned under the subheading “Face Validity Check and Piloting Phase”, a preliminary investigation was undertaken with a sample of individuals from the general population. The pilot study of the questionnaire confirmed several aspects, including the clarity and comprehensibility of the survey items, their alignment and suitability with the research objectives, the adequacy of the questionnaire's length and completion time, and the appropriateness of the response options and scale employed, see page 5, lines 138-146. Medical terms were translated to Arabic to enhance participants understanding.
- In Table 4, you asked: "Have you ever taken cortisone treatment (corticosteroid)?" Did you take this into account when it comes to your results? Perhaps folks who have taken the medication before are therefore more likely to know more about the medication than others, and so once again, this could skew your results.
- Thank you for this comment. This is one possibility, however, many patients use corticosteroids without having appropriate knowledge about this class of medications, which leads to misuse as we highlighted now in the introduction, see page 3, lines 86-89.
- Along these same lines, there is no delineation between the results as it comes to the general public vs. healthcare providers vs. what students were in university for, those who are furthering their education in the medical profession may have more knowledge.
- Thank you for this comment. We have had addressed this point by conducting logistic regression analysis to identify participants group who are more likely to use corticosteroids. We have now clarified further that the aim of the study was to assess levels of public awareness concerning corticosteroid use and side effects and predictors of its use not to examine the difference in the knowledge among different populations.
- Lines 171 - 175 - The section called Predictors of corticosteroids, please see my comments below; you cannot make this kind of claim when there are many other factors that need to be considered.
- Thank you for this comment. In this section, we conducted binary logistic regression analysis to identify predictors of using corticosteroids, which was one of our objectives.
Discussion
- Line 191 - You make the claim that "these findings imply that study participants have a good knowledge of corticosteroid administration methods..." What defines good? What are you comparing it to?
- Thank you for this comment. As you can see in the results section, most of the patients were able to identify corticosteroids dosage forms, indications of use, and side effects, see pages 7 and 8. No comparison was made as highlighted above, we aimed to examine the public awareness and identify predictors of their use.
- Line 195 - What is a chest disease?
- Thank you for this comment. We are sorry for this unintended mistake, we have now corrected it to be asthma, see line 226.
- Line 205 - You state "In contrast to our research findings, an earlier study found that pharmacists exhibited knowledge gaps regarding administration..." What does this have to do with your research findings when you never indicated that you had pharmacists in your cohort or that you looked specifically at pharmacist results?
- Thank you for this comment. We are sorry for this unintended mistake; we have now removed this sentence.
- Line 297 - You state "Furthermore, this finding suggests that improving access to medical care among this group could help reduce steroid phobia..." And yet there is no question regarding this in your questionnaire.
- Thank you for this comment. We agree with the reviewer on this point, but we preferred to mention it as there is a common corticosteroids phobia in our community; which could justify why older group did not have higher likelihood of using corticosteroids (higher odds ratio).
I appreciate the efforts that it took to complete this study - you had a large number of participants - however, there are several methodological issues that prevent me from recommending this paper to proceed.
- We would like to thank the reviewer for the time and efforts in reviewing our manuscript and for providing these comments. We have now addressed and clarified the comments mentioned by the reviewer and updated the manuscript accordingly. We will conduct further research in the future to investigate the points that were not covered in the current study. As you know, research is a continuous journey and we will continue working on it. We totally appreciate your understanding.
Reviewer 2 Report
Thank you for the opportunity to review this paper. It is an interesting study on an important topic. However, please address the following in order for it to be suitable for publication:
1. The study population is not clear. The title of the manuscript states Riyadh but this restriction is not listed in the methods, where it seems like the sample is drawn from the entire Saudi Arabian population. This needs to be clarified.
2. More information is needed on how the demographics of the study sample compare to the demographics of the target population to determine if this is a representative sample. Please add this information.
3. The introduction could be streamlined. For example, is the mechanism of action of corticosteroids and the history of their discovery and use necessary to the research you present?
4. Please explain what you mean by and provide a reference for your statement that corticosteroids are being misused by patients and physicians in your introduction.
5. Please check for grammatical, spelling and typographical errors, e.g. in lines 80-81 on page 2, this should be the co-authors conducted data collection through the utilisation... (or the co-authors collected data using an online questionnaire...)
6. Please ensure that all necessary information is provided for each reference in your reference list. Some references such as reference 2 seem to be missing information.
I look forward to reading the next version of this paper!
Authors to please check for grammatical, spelling and typographical errors, e.g. in lines 80-81 on page 2, this should be the co-authors conducted data collection through the utilisation... (or the co-authors collected data using an online questionnaire...)
Author Response
Reviewer 2:
Thank you for the opportunity to review this paper. It is an interesting study on an important topic. However, please address the following in order for it to be suitable for publication:
- The study population is not clear. The title of the manuscript states Riyadh but this restriction is not listed in the methods, where it seems like the sample is drawn from the entire Saudi Arabian population. This needs to be clarified.
- Thank you for this comment. We are sorry for this confusion, we have now re-wrote the title of the study to address the reviewer comment to be “Public awareness level regarding corticosteroid use and side effects: A cross-sectional study among user and non-user in Riyadh, Saudi Arabia”.
- More information is needed on how the demographics of the study sample compare to the demographics of the target population to determine if this is a representative sample. Please add this information.
- Thank you for this comment. We have now addressed this comment and added this information to the limitations of the study, see lines 344-352.
- The introduction could be streamlined. For example, is the mechanism of action of corticosteroids and the history of their discovery and use necessary to the research you present?
- Thank you for this comment. We have now addressed this comment and removed this information from the introduction.
- Please explain what you mean by and provide a reference for your statement that corticosteroids are being misused by patients and physicians in your introduction.
- Thank you for this comment. We have now addressed this comment and clarified this point, see lines 74-87.
- Please check for grammatical, spelling and typographical errors, e.g. in lines 80-81 on page 2, this should be the co-authors conducted data collection through the utilisation... (or the co-authors collected data using an online questionnaire...)
- Thank you for this comment. We have now addressed this comment.
- Please ensure that all necessary information is provided for each reference in your reference list. Some references such as reference 2 seem to be missing information.
- Thank you for this comment. We have now addressed this comment.
Round 2
Reviewer 1 Report
I thank the authors for their corrections and updates!
Author Response
Thank you for confirming that you are happy with this manuscript
Reviewer 2 Report
Thank you for revising this paper so quickly. All of my queries have been addressed and I feel this paper is now in a suitable form for publication.
English language is fine.
Author Response
Thank you for confirming that you have no further comments